# An exploration of the use of 3D printed foot models and simulated foot lesions to supplement scalpel skill training in undergraduate podiatry students: A multiple method study

**Helen A. Banwell**[1]*, **Ryan S. Causby**[1], **Alyson J. Crozier**[1,2], **Brendan Nettle**[1], **Carolyn Murray**[1]

1 Allied Health and Human Performance, University of South Australia, Adelaide, South Australia, Australia,
2 Alliance for Research in Exercise, Nutrition and Activity, University of South Australia, Adelaide, South Australia, Australia

* helen.banwell@unisa.edu.au

**Data Availability Statement:** All relevant data are within the paper and its Supporting information files.

## Abstract

### Background

Podiatrists regularly use scalpels in the management of foot pathologies, yet the teaching and learning of these skills can be challenging. The use of 3D printed foot models presents an opportunity for podiatry students to practice their scalpel skills in a relatively safe, controlled risk setting, potentially increasing confidence and reducing associated anxiety. This study evaluated the use of 3D printed foot models on podiatry students' anxiety and confidence levels and explored the fidelity of using 3D foot models as a teaching methodology.

### Materials and methods

Multiple study designs were used. A repeated measure trial evaluated the effects of a 3D printed foot model on anxiety and confidence in two student groups: novice users in their second year of podiatry studies (n = 24), and more experienced fourth year students completing a workshop on ulcer management (n = 15). A randomised controlled trial compared the use of the 3D printed foot models (n = 12) to standard teaching methods (n = 15) on students' anxiety and confidence in second year students. Finally, a focus group was conducted (n = 5) to explore final year student's perceptions of the fidelity of the foot ulcer models in their studies.

### Results

The use of 3D printed foot models increased both novice and more experienced users' self-confidence and task self-efficacy; however, cognitive and somatic anxiety was only reduced in the experienced users. All changes were considered large effects. In comparison to standard teaching methods, the use of 3D printed foot models had similar decreases in anxiety

**Funding:** the authors received no specific funding for this work.

**Competing interests:** The authors have declared that no competing interests exist.

and increases in confidence measures. Students also identified the use of 3D foot models for the learning of scalpel skills as 'authentic' and 'lifelike' and led to enhanced confidence prior to assessment of skills in more high-risk situations.

## Conclusion

Podiatry undergraduate programs should consider using 3D printed foot models as a teaching method to improve students' confidence and reduce their anxiety when using scalpels, especially in instances where face-to-face teaching is not possible (e.g., pandemic related restrictions on face-to-face teaching).

## Introduction

Podiatrists regularly use scalpel blades (scalpels) in the management of foot pathologies, such as for the removal of callus, corns, management of ingrown toenails and debridement of foot ulcers. As such, scalpel skills are a rudimentary part of the tertiary training for students studying to become a podiatrist [1]. Whilst each Australian University has slightly varying techniques, training of scalpel skills consistently involves a demonstration of scalpel use, with students then given supervised practice using inanimate objects (e.g., soap, oranges) prior to moving onto clients of student-based podiatry services [1]. Given that people with foot ulcers are more likely to be at higher risk of amputation if mis-managed, foot ulcer debridement practice is often limited to placements within high-risk foot clinics (e.g., clinics dedicated to foot ulcer/wound management and lower limb salvage based in tertiary health centres). An increase in student intake, a decrease in demand for university-based podiatry services or a reduction in available high-risk placements can therefore impact on student exposure to, and practice of, these critical skills.

The requirement for competent scalpel skills in Australian podiatry graduates has never been higher. Our ageing population has an increased prevalence of painful foot lesions, including callus and corns [2], which when managed well (including scalpel debridement) can assist in reducing pain [3], ultimately improving quality of life [4]. Furthermore, the non-traumatic lower limb amputation rate has risen 30% in the last ten years [5], with diabetes-related foot disease resulting in 4,400 lower extremity amputations and 1,700 deaths every year [6, 7]. Foot ulcers precede 84% of these amputations [8], and despite their management being multifactorial, scalpel debridement of callus that develops over foot ulcers due to the (thicker) skin physiology is considered one of the most effective [9].

Yet the teaching and learning of scalpel skills is challenging, there is inherent anxiety for students due to concerns for their own and client safety, intensified in 'high-risk' settings [1, 10]. Anxiety is known to be detrimental to the learning process [11, 12] particularly during the early stages of learning [13, 14] and highly demanding tasks [12, 15]. Confidence is a major predictor of anxiety, with increased levels of confidence reducing the effect anxiety has on performance [16, 17]. Novice podiatry students have previously demonstrated significantly higher levels of anxiety, and lower levels of confidence, than more experienced peers [18]. Bandura's [16] self-efficacy model suggests providing students with opportunities to be successful, without large consequences can increase an individual's confidence. Previous research has noted improved confidence [19] and accuracy in medical school residents using 3-dimensional (3D) printed models to detect fractures [20]. As such, providing podiatry students an opportunity

to practice their scalpel skills in relatively safe, controlled environments using 3D printed foot models may assist to mitigate these concerns.

This study aimed to evaluate the impact of using 3D printed foot models on podiatry students' confidence and anxiety levels and compare the use of 3D printed models versus traditional teaching methods. It was hypothesised that students would experience (1) an increase in confidence and reduction in anxiety after using 3D printed foot models, and (2) greater reductions in anxiety and gains in confidence when using 3D printed foot models compared to standard teaching methods. A secondary, more exploratory, aim was to determine student opinion on the fidelity of using 3D foot models (with added ulcers) through qualitative interviews.

## Materials and methods

### Study design

Given the multiple aims of this research, multiple study design methods were used [21].

A repeated measure trial evaluated the effect of 3D printed foot models on confidence and anxiety in two groups; second year students who were using a scalpel for the first time and fourth (final) year students who had participated in a half-day workshop on foot ulcer management. A randomised control trial (RCT) compared the use of 3D printed foot models to standardised teaching on confidence and anxiety in second years who were using a scalpel for the first time. A qualitative descriptive methodology [22] was used to explore our secondary aim; final year students' perceptions of the fidelity of the foot ulcer models. Students participated in a focus group, which allowed discussions around characteristics, traits, and behaviours that occur in everyday context using common language to occur [22], (Table 1).

Ethical clearance for this project was obtained by the Human Research Ethics Committee of the University of South Australia (Approval number 201908). All participants provided written informed consent prior to enrolment.

### Participants

Participants were sought via purposive sampling of undergraduate podiatry students enrolled at the University of South Australia in the second year of the course in 2019 and 2020, and final year students enrolled in 2019. There were three groups of students enrolled: second year (2019 cohort), second year (2020 cohort) and final year students. A subset of the final year students participated in the focus group (Table 1).

**Table 1. Participant characteristics for the multiple method studies.**

| Study aim | Study design | Participant group/s | Exposure to 3D printed foot models | n | Age in years (M ± SD) | Gender (M:F) |
|---|---|---|---|---|---|---|
| Effect on anxiety and confidence | Repeated measure | Final year students | 1 x half-day foot ulcer management workshop | 15 | 23.5 ± 1.8 | 6:9 |
| | | Second year students (2020 cohort) | 1-hour training and six-weeks self-paced use | 24 | 22.7 ± 5.8 | 9:15 |
| Comparison to standard teaching | RCT | Second year students (2019 cohort) | Nil (control) | 15 | 22.1 ± 3.8 | 6:9 |
| | | | 1-hour training (intervention) | 12 | 20.6 ± 1.7 | 5:7 |
| Fidelity of models | Focus group | Final year students | 1 x half-day foot ulcer management workshop | 5* | NR | 1:4 |

*participants were a subset of the final year student cohort that completed the foot ulcer management workshop.

RCT—randomised control trial, NR—not recorded.

**Effect on anxiety and confidence.** A repeated measure study identified changes in anxiety and confidence for two groups; final year podiatry students (n = 15) who attended a half-day foot ulcer management workshop, and second year students (2020 cohort), (n = 24) that had not previously used a scalpel. Due to placements occurring across the year, final year students may or may not have completed a placement that included debridement of foot ulcers.

**Comparison to standard teaching.** A randomised control trial compared standard teaching (control) to teaching using a 3D printed foot model (intervention) for changes in anxiety and confidence in second year students (2019 cohort) who were using a scalpel for the first time. Students received 1-hour of training after they were randomly allocated to the control (n = 15) or intervention group (n = 12).

**Fidelity of models'.** A small group of final year students (n = 5) who completed the foot ulcer management training were invited separately for a focus group via email from a single researcher (CM). While the focus group was an informal discussion, CM directed some discussion via an interview guide (S1 Appendix). These participants were also involved in the repeated measure study. To ensure participants were assured of anonymity, focus group participants characteristics (name, gender, age, experience) were not collected and researchers involved with the development of the foot models, teaching of scalpel skills and the analysis of the repeated measure study (HB, RC, BN, AC) were excluded from involvement in, or analysis of the focus group, remaining blinded to focus group participants. A $50 gift card was available to those who participated in the focus group to compensate for their time and efforts.

Students suffering physical disability, or taking medications likely to affect hand function, were excluded via existing criteria for mandatory student registration with the Podiatry Board of Australia.

## Intervention

There were two versions of 3D printed foot models used. A flexible Foot ulcer model with appliable lesions for final year students to use in foot ulcer management training (Fig 1), and a more rigid, robust Callused foot model for second year students learning scalpel skills for the first time (Fig 2).

The Foot ulcer models are printed in Ninja flex® filament to have adequate flex (to mimic foot motion) whilst the Callused 3D models are printed in standard nylon filament to be sufficiently robust (to withstand efforts from novice scalpel users). Both models were printed with three moulded divots on the bottom of the foot and two small circular divots on the top of the toes to represent foot ulcers or callus and corns. Callus and corns were produced using Flexible Polyurethane Resin (F-140), (AMC, Edwardstown, Adelaide).

The Foot ulcer models include a simulated 'exposed tendon' under the fifth metatarsal area (Fig 1). Ulcers are applied using a combination of commercial grade 'body ooze' (Barnes, Moorebank, NSW), to simulate blood and exudate, and cake frosting to mimic macerated wound tissue. Wounds are covered by the same flexible resin as used in callus, which is lanced by students during training (Fig 3).

## Outcome measures

As there is no valid or reliable measure of anxiety or confidence specific to podiatry skills, outcomes were measured using a modified existing tool and a purpose-built questionnaire.

**Anxiety and self-confidence.** Anxiety and Self-confidence was measured using the Competitive State Anxiety Inventory-2 (CSAI-2) [23], with wording modified to represent the task of using a scalpel. For example, the original item "I am concerned I may not do as well in this competition as I could" was modified to read "I am concerned I may not do as well using a

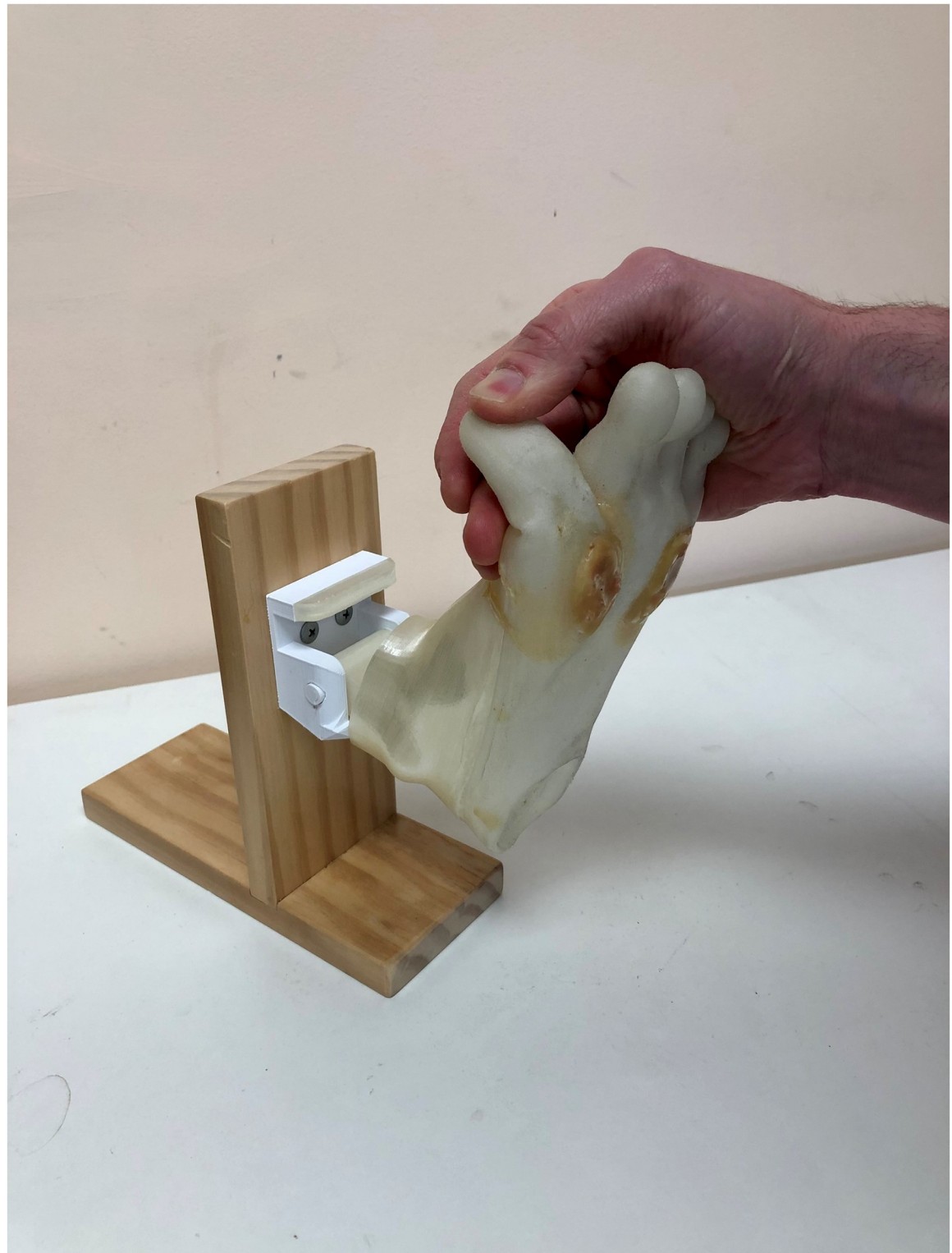

**Fig 1. Foot ulcer model.**

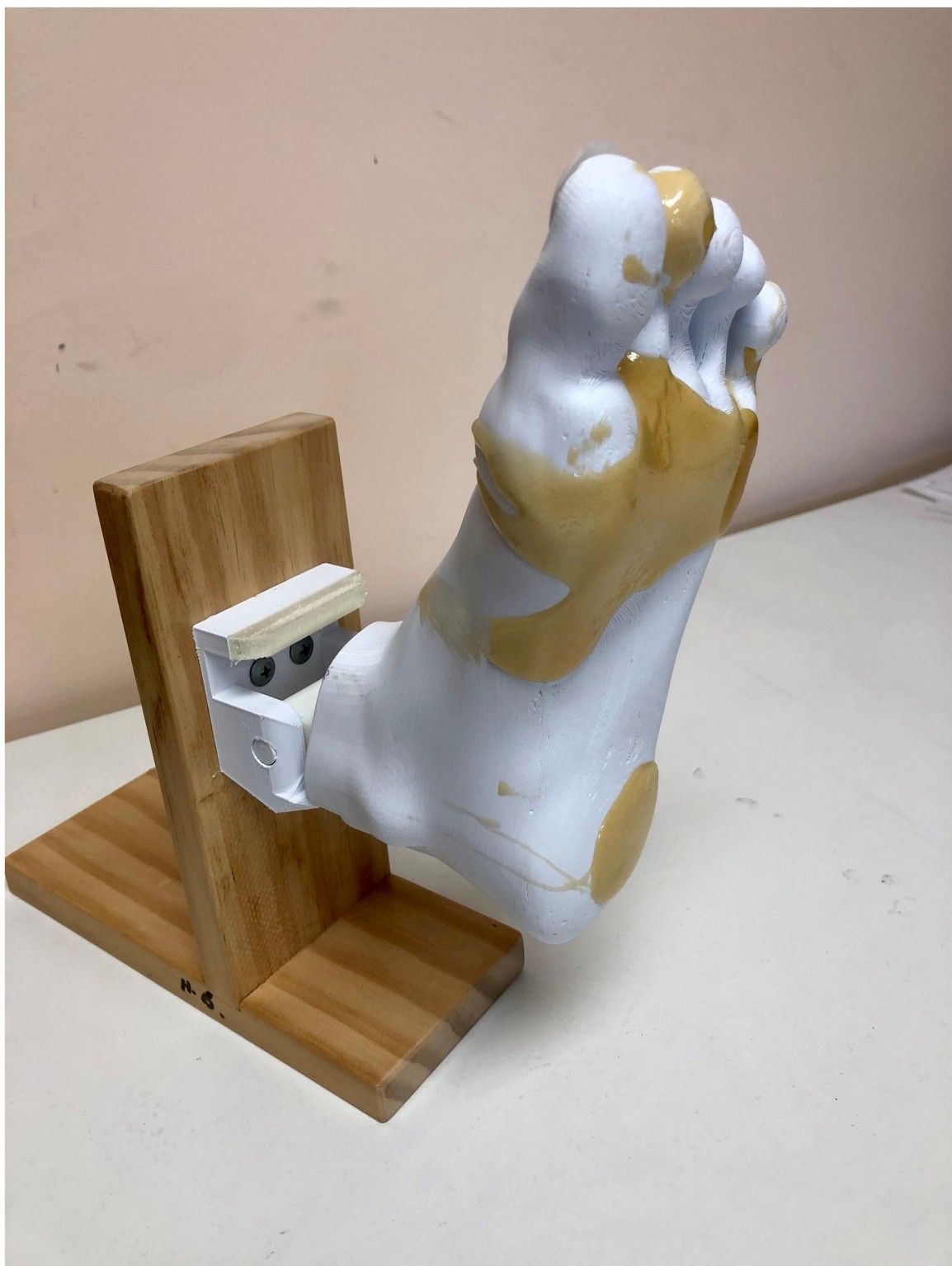

**Fig 2. Callused foot model.**

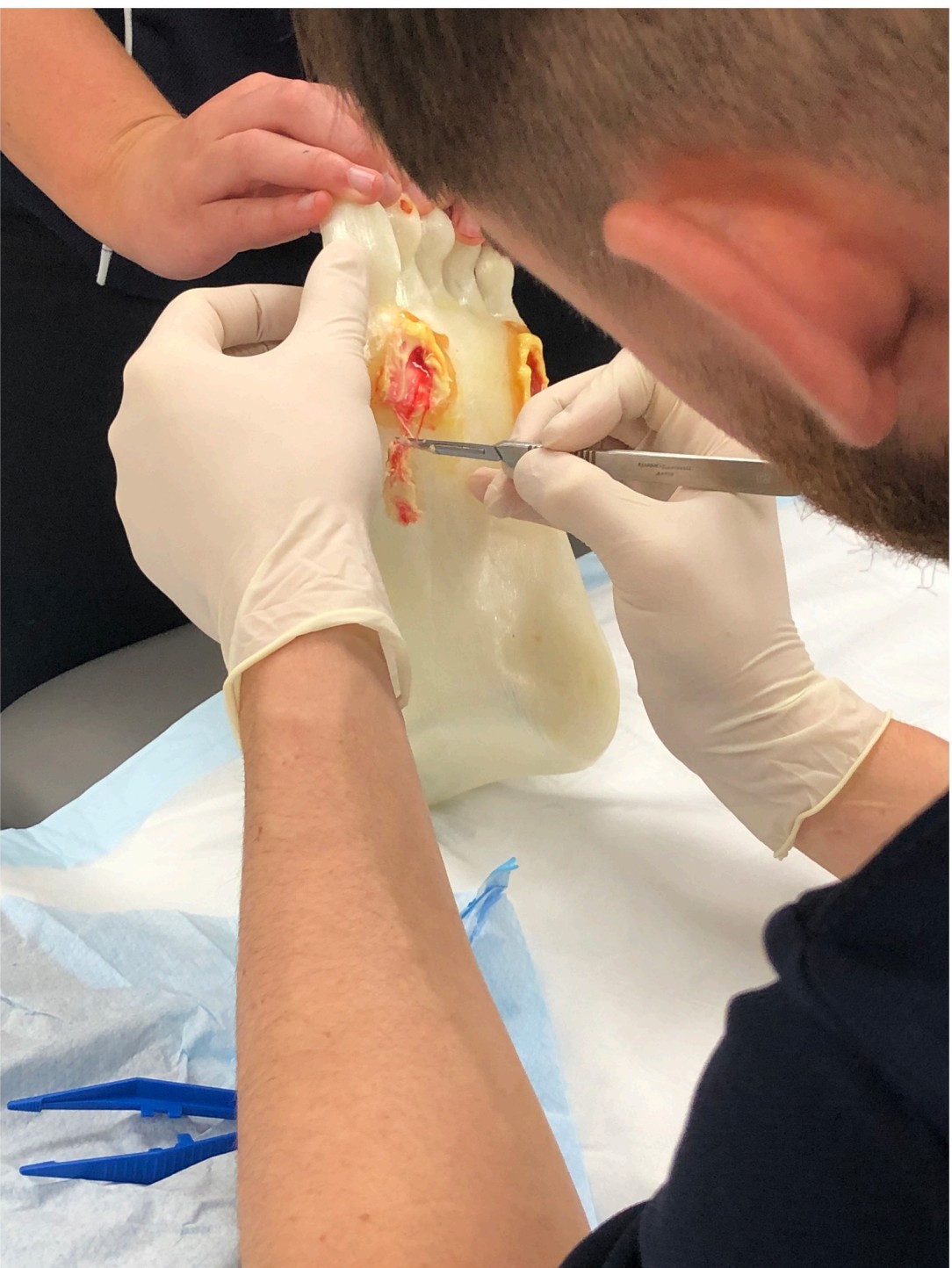

**Fig 3. Foot ulcer debridement.**

scalpel as I could" (S2 Appendix). The CSAI-2 includes 27 questions, across three subscales: cognitive state anxiety, somatic state anxiety, and self-confidence. Each subscale includes nine items, which are summed to represent the level of intensity the student is feeling. Participants were instructed to report their feelings in the present moment, just before using a scalpel.

Items were responded to on a four-point Likert scale response ranging from 1 (*not at all*) to 4 (*very much so*). The CSAI-2 has been shown to be a valid and reliable tool in a sport setting [24] and was found to be internally consistent in the present study (Cronbach alpha (self-confidence domain) = 0.75).

**Task self-efficacy.**   Students' confidence in their ability to use a scalpel was measured using a purpose-built questionnaire (S2 Appendix). Seven questions measured students' confidence specific to outcomes deemed important to podiatry educators. An example item included "How confident are you in your ability to stabilise your hand when using a scalpel?". Items were scored on a VAS scale (0–100 mm) where 0 represented no confidence at all and 100 represented as confident as you've ever felt (see S2 Appendix for all items). An average task self-efficacy score was calculated from the seven questions, with the items showing adequate internal reliability (Cronbach alpha = 0.94).

## Procedure

All potentially eligible participants were alerted to the various studies via e-mail two weeks prior to the relevant undergraduate-level courses being conducted, and in person during the course introduction sessions. Potential participants were given written information regarding the study and advised their involvement was voluntary and that they could withdraw at any time without consequence.

**Effect on anxiety and confidence.**   The two groups involved in this study had different protocols to suit the teaching requirements.

Final year students (n = 15) were introduced to the Foot ulcer models (Figs 2 & 3) during a foot ulcer management workshop held in August 2019. The workshop included a review of foot ulcer management theory, with the 3D printed foot ulcer models used in classification and sizing of wounds, cleaning, identification, and application of appropriate dressings as well as the debridement of the foot ulcers. Students worked in pairs, helping to stabilise the model and record measures for each other, and were instructed to observe and adhere to aseptic techniques and infection control guidelines. Participants completed measures of anxiety, self-confidence and task self-efficacy immediately prior to the workshop (baseline) and immediately after (follow-up).

Second year students (2020 cohort), (n = 24) were introduced to the Callused 3D foot models (Fig 1) for scalpel skills teaching in small groups (n = ~8) in April 2020, each receiving their own model and given a 1-hour training session with an experienced clinical tutor. They were then allowed unlimited self-paced practice over a six-week block. Importantly, this training and use occurred during COVID-19 related restrictions on face-to-face teaching, therefore all models and instruments were supplied by postal services prior to the training session, with training conducted via web-based telecommunications. Participants completed the measures of anxiety, self-confidence, and task self-efficacy immediately prior to the workshop (baseline) and after six-weeks of self-paced use (follow-up).

**Comparisons to standard teaching.**   Second year students (2019 cohort) were randomised into two groups using a computer-generated randomisation schedule (https://www.randomizer.org): the control group (standardised teaching, n = 15) or the intervention group (Callused 3D foot models, n = 12). The standardised teaching group were introduced to scalpel skills training in three small groups (n = ~5) and given a 1-hour training session with an experienced clinical tutor using a scalpel to 'debride' a bar of soap. The intervention group was introduced to the Callused 3D foot models in two small groups (n = ~6) and given a 1-hour training session with an experienced clinical tutor using the 3D foot models 'debriding' the applied 'callus and corn' lesions. Participants completed measures of anxiety, self-confidence

and task self-efficacy immediately prior to the training (baseline) and immediately after the training (follow-up).

**Fidelity of models'.**   A subset of final year students who had participated in the foot ulcer management group (n = 5) also participated in a focus group, which was conducted at with an expert in qualitative research (CM) in a private room at the university. Questions of the focus group were designed by the research team with the aim of gathering students' perspectives on the feasibility and effectiveness of using the 3D printed foot ulcer models for foot ulcer management training. A semi-structured focus group guide was developed, with questions being open-ended. The focus group lasted for 60 minutes, was recorded and professionally transcribed.

## Data analysis

Descriptive statistics were used to describe participant characteristics at baseline. Where data were missing for one or more outcomes, participants were excluded from all analysis. Data analysis for the quantitative trials were conducted in IBM SPSS 21 (IBM Corp, 2012, Armonk NY, USA). CSAI-2 outcomes (in points) for the three subscales (cognitive anxiety, somatic anxiety, self-confidence) were summed as instructed [24], with the seven items of the task self-efficacy questionnaire (VAS, mm) aggregated to display group means (range 0 to 100). Normality of data were assessed using Shapiro-Wilks tests ($p \geq 0.05$), (S3 Appendix). To determine intervention effects, *Cohen's d* were calculated and interpreted based on Cohen guidelines [25], where small effect $\geq 0.2$, medium effect $\geq 0.5$ and large effect $\geq 0.8$. Effects less than 0.2 were considered very small. Statistical significance was set at $p \leq 0.05$.

To determine the effect on anxiety and confidence a repeated measure design investigated differences between baseline and follow up for anxiety, self-confidence and task self-efficacy using a within-subject paired t-tests (two-tailed) where data were normally distributed. Where outcomes were not normally distributed, data was analysed using raw and transformed ($Log^{10}$) data. If results using the transformed data were not different from those using the raw data, the raw data is reported for ease of interpretation.

To examine whether anxiety, self-confidence and task self-efficacy differed between standard teaching and the use of the models (i.e. control vs. intervention), a repeated-measures analysis of variance was conducted. No post hoc tests were applied.

To explore the fidelity of the models a transcript from the focus group was analysed using the six phases of reflexive thematic analysis [26]. After reading the transcript in detail, the researcher undertook line by line open coding of the transcript followed by an analysis of the codes for patterns and consistencies. This process resulted in development of four themes that describe the findings from the focus group.

Due to the recruitment requirements limiting the recruitment sample to students enrolled in undergraduate podiatry courses during 2019 or 2020, and a novel intervention where effect size could not be assumed, an *a priori* sample size calculation was not conducted. A *post-hoc* power calculation was conducted based on outcomes from the self-confidence domain of the CSAI-2 and using GPower3 (two-tailed t-tests, power (1 - β) set and α = 05) [27].

## Results

### Participant characteristics

A total of 66 participants participated across the multiple studies (Table 1). The majority were female (61%), with a mean age of 22.2 (SD 3.8) years. Three participants did not complete the CSAI data, they are included for Task Confidence outcomes alone (Tables 2 and 3).

**Table 2. Repeated measure study comparing anxiety and confidence (using the CSAI-2 and purpose-built questionnaire (VAS)) outcomes prior (baseline) and following (follow up) a half-day foot ulcer management workshop using 3D printed foot ulcer models in final year students.**

| Participant group | Measure | n | Time point | Mean | SD | d | p |
|---|---|---|---|---|---|---|---|
| Final year students | Cognitive state anxiety | 14 | Baseline | 17.71 | 5.99 | | |
| | | | Follow up | 15.07 | 4.80 | 1.45 | 0.02 |
| | Somatic state anxiety | 14 | Baseline | 15.79 | 4.25 | | |
| | | | Follow up | 12.93 | 1.90 | 1.63 | 0.01 |
| | Self-confidence | 14 | Baseline | 22.07 | 4.12 | | |
| | | | Follow up | 27.29 | 8.46 | 1.60 | 0.01 |
| | Task self-efficacy (VAS) | 15 | Baseline | 65.71 | 16.34 | | |
| | | | Follow up | 77.78 | 13.74 | 2.71 | 0.00 |

3D = three-dimensional, CSAI-2 = competitive state anxiety inventory-2, VAS = visual analogue scale

**Effect on anxiety and confidence.** For final year students (n = 15), a significant decrease was observed in cognitive state (t (13) = 2.62, p = 0.02) and somatic state anxiety (t (13) = 2.94, p = 0.01) from baseline to follow-up. Cognitive state anxiety reduced an average of 2.64 points (SD = 3.77; 95% CI [0.46, 4.82]) while somatic state anxiety decreased by 2.86 points (SD = 3.63; 95% CI [0.76, 4.95]), with large effect sizes (1.45 and 1.63) respectively (Table 2). In addition, a statistically significant increase was observed for self-confidence (t (13) = 2.88, p = 0.01) and task self-efficacy (t (14) = 5.07, p = 0.00). Self-confidence increased by an average of 5.21 points (SD = 6.78; 95% CI [1.29, 9.13]), with task self-efficacy improved by a mean of 12.07 mm (SD = 9.22; 95% CI [6.97, 17.18]). Cohen's d calculations indicated a large effect size change (self-confidence = 1.60; task self-efficacy = 2.71), (Table 2).

For second-years students (2020 cohort) no significant difference was observed in cognitive anxiety or somatic anxiety from baseline to follow-up (Table 3). There was a statistically significant increase in students' self-confidence (t (21) = 3.66, p = 0.00) and task self-efficacy (t (23) = 6.23, p = 0.00) following six weeks of using the 3D foot model. Self-confidence increased by an average of 3.86 points (SD = 4.95; 95% CI [1.67, 6.06]), and task self-efficacy increased an average of 30.48 mm (SD 23.97; 95% CI [20.35, 40.60]). Again, large effect size changes were observed (self-confidence = 1.59; self-efficacy = 2.60).

The outcomes were sufficiently powered for final year students (d = 0.71, 1.72 (21), power = 0.94) and second year students 2020 (d = 0.75, 1.71 (24), power = 0.96) respectively.

**Table 3. Repeated measure study comparing anxiety and confidence (using the CSAI-2 and purpose-built questionnaire (VAS)) outcomes prior (baseline) and following (follow up) 1-hour training and six-weeks self-paced use in second year students (2020 cohort).**

| Participant group | Measure | n | Time point | Mean | SD | d | p |
|---|---|---|---|---|---|---|---|
| Second year students 2020 | Cognitive state anxiety | 22 | Baseline | 16.68 | 3.94 | | |
| | | | Follow up | 16.50 | 3.91 | 0.11 | 0.80 |
| | Somatic state anxiety | 22 | Baseline | 15.23 | 3.79 | | |
| | | | Follow up | 13.95 | 2.48 | 0.62 | 0.17 |
| | Self-confidence | 22 | Baseline | 23.09 | 5.78 | | |
| | | | Follow up | 26.95 | 4.28 | 1.59 | 0.00 |
| | Task self-efficacy (VAS) | 24 | Baseline | 43.33 | 23.66 | | |
| | | | Follow up | 73.80 | 12.37 | 2.60 | 0.00 |

3D = three-dimensional, CSAI-2 = competitive state anxiety inventory-2, VAS = visual analogue scale

**Table 4. Randomised control trial comparing anxiety and confidence (using the CSAI-2 and purpose-built questionnaire (VAS)) outcomes for standard teaching and the use of 3D printed foot models for 1-hr in second year students (2019 cohort).**

| Measure | Time | Standard teaching | | | | 3D foot model group | | | | p |
|---|---|---|---|---|---|---|---|---|---|---|
| | | n | M | SD | d | n | M | SD | d | |
| Cognitive state anxiety | Baseline | 14 | 16.85 | 4.12 | 0.34 | 12 | 17.67 | 6.97 | 0.13 | 0.85 |
| | Follow up | | 18.50 | 5.45 | | | 18.41 | 4.58 | | |
| Somatic state anxiety | Baseline | 14 | 14.85 | 2.38 | 0.89 | 12 | 16.16 | 6.53 | 0.10 | 0.87 |
| | Follow up | | 17.50 | 3.48 | | | 16.66 | 3.20 | | |
| Self-confidence | Baseline | 14 | 22.71 | 5.64 | -0.35 | 12 | 17.92 | 8.45 | 0.55 | 0.48 |
| | Follow up | | 20.71 | 5.64 | | | 22.25 | 7.20 | | |
| Task self-efficacy (VAS) | Baseline | 15 | 46.28 | 23.70 | 0.87 | 12 | 49.05 | 29.87 | 0.68 | 0.79 |
| | Follow up | | 64.12 | 16.82 | | | 65.48 | 16.55 | | |

3D = three-dimensional, CSAI-2 = Competitive State Anxiety Inventory-2, VAS = visual analogue scale

**Comparisons to standard teaching.** For second year students (2019 cohort), no significant interaction effect was observed for cognitive state anxiety (Wilks' Lambda = 0.99, $F$ (1, 24) = 0.20, p = 0.66), somatic state anxiety (Wilks' Lambda = 0.94, $F$ (1, 24) = 1.61, p = 0.22) or task self-efficacy (Wilks' Lambda = 0.99, $F$ (1, 25) = 0.03, p = 0.85). There was a significant interaction effect for self-confidence over time for the control and intervention group (Wilks' Lambda = 0.83, $F$ (1, 24) = 5.04, p = 0.03), suggesting most participants improved their general self-confidence through training. No group interaction was observed, however, such that those who were exposed to the standard teaching method and the 3D foot models experienced similar anxiety, self-confidence, and task self-efficacy (Table 4). Non-significant effect size change ranged from very small (0.10) to large (0.89). This study was underpowered ($d$ = 0.23, 1.71 (24), power = 0.15).

**Fidelity of models'.** Four themes were identified on perceived and recommended fidelity following analysis of responses from final year students.

1. Models were lifelike

   Students reported contradictory information about the feel of the models and comparisons made with the feel of a real foot. Those who said the models felt '*hard*' said this supported their learning because it made it easier to practice anchoring their hand to debride. In contrast, for some, the models felt too soft, although they appreciated the in-between step before working on a 'real' foot.

   "*So, it's good to have something like a step in between that, so you're still not on a real person, like, using a real person, but it is quite realistic to what a real foot would be like. It's a good in-between and to progress to develop the skills . . . compared to what we've had before . . . but it was very soft*"

   Students reported that the skill of anchoring was key, and the models were optimal for learning this. Although they regarded the models as more '*lifelike*' than other models they had used; they were "*never going to be the real thing, bottom line*". Working on the models when they were not connected to a body was described as challenging "*because it's not in that fixed position . . .it's really hard to put it into a position to debride in the right area. . .we had to figure that out ourselves*". The students believed that it should be possible to connect the 3D printed foot to something when working on them in class: "*The really hard thing*

*is . . . when you're holding a 3D (printed) model, it's just a foot and you're trying to sort of put some tensile stress onto the foot. It's not connected to anything."*

2. Replicating the ulcer

They recommended more contrasting colours while learning as otherwise they found it confusing about what was skin and what was callus. They did not feel it was an issue that the model was not entirely authentic in terms of colouring as it was more important to be able to clearly distinguish the different parts of the foot and ulcer, which was enabled with the darker colour models. They liked how each foot was different as that reflected what occurs in practice with everyone getting a unique experience. The ulcers themselves were described as *"a bit rubbery"*, but *"well done"* because they simulated a real ulcer authentically with blood, exudate and layers.

*"It'd be very hard to replicate a real ulcer, but I think the fact they had almost different layers and tissue and that's literally what you get in real life. You just have dead tissue, new tissue coming through. Sometimes there'll be blood."*

Students identified that sometimes during the process of cleaning, the students were too vigorous, and if the ulcer was too soft, some parts of the model would come off that would not normally. Also, the students advised that replicating an ulcer with odour would help prepare them for that experience.

*"I found that a struggle with the first ulcer, high-risk ulcer I saw. I wasn't even debriding it, but the smell is what got to me the most. I think if they can integrate that into the simulation, that's good"*

3. Previous exposure affected experience

Those who had experienced a relevant placement and been exposed to foot ulcers already, identified the 3D printed models to be less important for their learning compared with those who had not seen a real foot with this pathology. They reflected about the contextual nuances that are not available when working with the models.

*"There's a lot going on, you're thinking about your bedside manner, you're thinking about what you have to do, you're taking in the smell, the look of it and you have to think about maintaining sterility. Yeah, there's a lot going on [with a real foot]."*

Those who were new to the experience reported taking the opportunity very seriously and valued the learning experience more highly than those who had seen ulcers on placement. There was a suggestion that the 3D printed models would be ideal for students in second and third year who have not been on placement yet. Those who were exposed to the models after seeing them on placement reflected that the models would have been good preparation for placement.

*"For me personally, I'd done all my placement before, so I didn't find that [3D printed models] helped at all. But it would have been really, really helpful maybe last year. It could have been amazing to have some kind of idea of what an ulcer would look like and how you about debriding it"*

4. Authenticity builds confidence

Having to debride an ulcer for a high-risk patient is *"most feared among students"*. It was acknowledged that replicating a high-risk patient would be challenging but being taught how to debride with the model was valued as it built their confidence. The opportunity was described as desensitising and students recommended replicating a clinical situation as much as possible to improve authenticity (e.g., using sterile gloves, using disposable trays). The students reflected that feeling confident in one aspect of the situation, meant they could focus on other aspects, rather than feeling overwhelmed by everything being new.

"*I think if you only take one thing from it, so just say you might take the sterilisation aspect from it, or I might only take the scalpel skills from it, I think, even if you're confident in that one area, when you're in a real-life situation, that can really, really help you.*"

They saw potential for using the 3D printed models to practice removing and applying dressings on ulcers, as even opening a dressing pack was new to many of them. Being able to see different approaches to debridement and then getting to continue practicing with the model at home was regarded as *"really cool"*. They recommended having plenty of tutors in the room to enable *"more eyes"* on what they were doing, opportunity to ask questions and sharing of stories from clinicians who work with these patients regularly. They also recommended smaller groups as they would take it more seriously and suggested adding some context such as linking the foot to a scenario involving a person's history so they could pretend it was a real person and role play subjective questioning.

## Discussion

The primary aim of this study was to evaluate the use of 3D printed foot models on podiatry student's anxiety and confidence when using scalpels, and to determine if using 3D printed models had any benefit over standard teaching practices. Further, we explored student's perception on the fidelity of using 3D printed models for foot ulcer management training. Results identified that the use of 3D printed models led to increases in novice and more experienced students' confidence levels, as well as reduced anxiety in the more experienced group. Importantly, the use of 3D printed models had similar increases in confidence and reductions in anxiety when compared to standard teaching methods in novice scalpel users. In addition, final year students provided mostly positive comments about the models' inclusion in the teaching regime, with students suggesting the use of 3D printed models specifically for foot ulcer management training should be incorporated earlier in their degree.

As hypothesised, the use of 3D printed models in training did decrease podiatry students' anxiety and increase their confidence. This finding supports predictions from Bandura's (1997) self-efficacy theory, such that providing successful experiences with a task in a relatively safe and controlled situation (3D printed foot model) improves general and task-specific confidence. This should improve the inherent anxiety associated with learning to use scalpels, and when using them on 'high-risk of amputation' populations. Consequently, the use of 3D printed feet provides an alternative teaching method to assist students in learning scalpel skills, and they do so in a manner which is safer, less anxiety provoking and one which requires fewer human resources (e.g., reduced supervision). These results also extend the findings with other 3D printed model research from medical residents' confidence to identify a fracture [19, 20], to undergraduate podiatry student's confidence to use scalpels with high-risk conditions (i.e., foot ulcers).

Interestingly, results seemed to differ for novice versus experienced students, with novice students only experiencing confidence boosts with no impact on their anxiety levels, whereas the final year students also experienced reductions in anxiety alongside increased confidence. This finding may reflect where the training sits for these groups within the course. In second-year, scalpel skill teaching forms part of an extensive pre-clinical module that culminates in students having to assess and manage 'genuine' podiatry clients for the first time. The knowledge that they may need to use scalpels on real people within a few weeks of training may cultivate a level of anxiety in second year students that was not modified with the use of 3D printed models. In contrast, this anxiety has been partially appeased by exposure to clinical practice in the more experienced students.

In contrast to expectations, for novice users using scalpels for the first time, the use of 3D models impacted confidence and anxiety similarly when compared to traditional teaching methods. In other words, 3D printed foot models and traditional face-to-face teaching methodologies performed equally as well to increase student's confidence in their abilities, and reduce their anxiety, when using scalpels. Which, whilst not supporting the use of 3D printed foot models over standard 'debridement teaching' (such as that conduced on soap or wax), these results provide evidence that the models are an effective replacement of traditional instruction methods. As this RCT study was only conducted with a 1-hour training session, examining the impact of having longer exposure and practice opportunities than occurred within this study, and investigating student preferences may assist in determining if value exists in one method of teaching over the other.

The qualitative results support the fidelity of authenticity in simulations for student learning outcomes [28]. Participants commented that it would be very hard to replicate a real ulcer and that it was never going to be the real thing, but even so, there were some elements that were seen as authentic including setting up the environment with expectations to sterilize and the feel of the model foot. Suggestions for enhancing authenticity including creating a teaching environment with increased tutor supervision, having to practice skills such as bedside manner and explaining what they are going to do to a simulated client and having the model fixated to resemble a foot and leg presentation more closely. Despite the challenges with creating authenticity, there were learning outcomes reported and students built their confidence. The students also made recommendations that the timing of the foot ulcer management experience would sit better if conducted earlier than final year in their four-year program. This recommendation was made in relation to timing of their placement experiences with the view that the simulation is valuable preparation. The relationship between simulation and how students perform on placement is under-reported in the literature [28]. However, there is some evidence that simulation enables educators to observe and assess students in a controlled environment before practicing with 'real' people on placements [28].

The students in final year also found value in practicing more than just 'debridement' process, as they spoke highly of the additional requirements (e.g., aseptic technique, cleaning of the wound etc.,) involved in the training, which may account for the reduced anxiety measured within this group. This bodes well, as changes in the Australian teaching and practice landscape require better preparation in foot ulcer management for all podiatry graduates. Previously considered a specialist skill set, graduates employed into high-risk foot clinics are often required to complete further training in-house prior to being able to manage clients independently. Those that are employed within the private podiatry services often refer clients with ulcers directly to these specialised clinics. However, the implementation of Chronic Disease Management (CDM) plans and the National Disability Insurance Scheme (NDIS) has impacted on this practice model. Podiatry provides the largest uptake of private allied health services under the CDM scheme [29] and strongly representative within the NDIS arena. This

has increased the exposure of private sector podiatrists to clients with chronic disease, disability, and the consequential active foot ulcers. Newly graduated podiatrists may work alone, without onsite mentorship, and given increasing demand on high-risk placements it is plausible that a person with a foot ulcer, already at extreme risk of amputation, may be managed by a podiatrist without clinical exposure to ulcer debridement. As tertiary education providers are experiencing significant financial obstacles from the impacts of COVID-19 and recent government funding reforms [30, 31], the use of 3D printed models may provide a cost-effective alternative to face-to-face delivery, ensuring training and practice of both early scalpel users and to ensure new graduates enjoyed improved ulcer management skills.

## Strengths & limitations

As 3D printed foot models had not, to the best of our knowledge, been investigated as a podiatry teaching resource previously, much of our investigations was explorative. The multiple study designs used allowed impact to be measured and compared across two cohorts of novice students and one cohort of final year students, tailored to allow data collection to continue through COVID-19 related campus closures and capturing as much quantitative and rich qualitative data as possible for a small cohort of participants. However, there are several limitations to acknowledge. Comparisons between the quantitative studies were limited by methodology and protocol differences, and the focus group also had a small sample size. Specifically, as the repeated measures methodology lacked control group comparisons, we are unable to determine if the measured anxiety and confidence improvements were due to the foot model exposure or related to time alone and comparisons cannot be made between this and the RCT findings. Different models and protocols were used between the second year and final year cohorts limiting the ability to make comparisons across the years. Furthermore, sample sizes, limited by available student numbers, were not consistent between the cohort groups. However, given the large effect size improvements observed and the positive thematic outcomes of focus group feedback, it is plausible to state that 3D printed models supplement clinical teaching well. This also infers that podiatry clinical teaching could be maintained during teaching interruptions, such as pandemic related restrictions on face-to-face teaching.

## Conclusion

Our findings indicate that exposure to 3D printed models is effective for increasing confidence in novice and experienced scalpel users, as well as and reducing anxiety among the more experienced students. It was also identified that 3D printed models were equally effective to traditional teaching styles for reducing anxiety and increasing confidence in novice users when given 1-hour of training. Positively, students reported the 3D printed models were an effective teaching modality, and they offered insights into how the models could be leveraged to further enhance students' learnings. Overall, the use of 3D printed foot models for use in tertiary podiatry education was supported.

## Supporting information

**S1 Appendix. Interview guide for focus group.**
(DOCX)

**S2 Appendix. Competitive State Anxiety Inventory-2 (CSAI-2) and purpose-built questionnaire (VAS).**
(DOCX)

**S3 Appendix. Raw data for quantitative studies.**
(DOCX)

## Author Contributions

**Conceptualization:** Helen A. Banwell, Carolyn Murray.

**Formal analysis:** Ryan S. Causby, Carolyn Murray.

**Investigation:** Helen A. Banwell, Brendan Nettle, Carolyn Murray.

**Methodology:** Helen A. Banwell, Ryan S. Causby, Alyson J. Crozier, Brendan Nettle.

**Validation:** Alyson J. Crozier.

**Writing – original draft:** Helen A. Banwell, Ryan S. Causby, Carolyn Murray.

**Writing – review & editing:** Alyson J. Crozier, Brendan Nettle.

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
