## [Decision Letter · Decision Letter 0]

11 Oct 2021

PONE-D-21-26441An exploration of the use of 3D printed foot models and simulated foot lesions to supplement scalpel skill training in undergraduate podiatry students: a multiple method study.PLOS ONE

Dear Dr. Banwell,

Thank you for submitting your manuscript to PLOS ONE. After careful consideration, we feel that it has merit but does not fully meet PLOS ONE’s publication criteria as it currently stands. Therefore, we invite you to submit a revised version of the manuscript that addresses the points raised during the review process.

Please address the comments of the reviewers, particularly review the method section.. Using a pictorial presentation of the different groups may make it easier for the reader. Alternately you could bring Table 1, earlier in the manuscript as it nicely summarises the groups.

We look forward to receiving your revised manuscript.

Kind regards,

Oathokwa Nkomazana, MD MSC PhD

Academic Editor

PLOS ONE

Additional Editor Comments (if provided):

Thank you for the very interesting study.

Please address the comments of the reviewers, particularly review the method section.. Using a pictorial presentation of the different groups may make it easier for the reader. Alternately you could bring Table 1, earlier in the manuscript as it nicely summarises the groups.

Journal Requirements:

2. Please include a copy of the interview guide used in the study, in both the original language and English, as Supporting Information, or include a citation if it has been published previously.

3. For more information on PLOS ONE's expectations for statistical reporting, please see https://journals.plos.org/plosone/s/submission-guidelines.#loc-statistical-reporting. Please update your Methods and Results sections accordingly.

Reviewers' comments:

Reviewer's Responses to Questions

**Comments to the Author**

1. Is the manuscript technically sound, and do the data support the conclusions?

Reviewer #1: Yes

Reviewer #2: Yes

2. Has the statistical analysis been performed appropriately and rigorously? 

Reviewer #1: Yes

Reviewer #2: Yes

3. Have the authors made all data underlying the findings in their manuscript fully available?

Reviewer #1: Yes

Reviewer #2: Yes

4. Is the manuscript presented in an intelligible fashion and written in standard English?

Reviewer #1: Yes

Reviewer #2: No

5. Review Comments to the Author

Reviewer #1: The aim of the study was to evaluate the use of 3D printed foot models on podiatry students anxiety and confidence when using scalpels and to determine if using 3D printed models had any benefit over standard teaching practices.

The paper is well written and structured, however a few questions arise.

- In the repeated measure trial and randomised controlled trail, sample size is varied and not kept constant. Is there a reason for not using equal sample size?

-On procedure, 4th years n =15 used models on fig 2 and 3 and worked in pairs, whereas 2nd years n =24 used model on figure 1 and worked in groups of about 8. Is there a reason why the two groups did not work on the same models as well as keeping consistency of the setting and either working in pairs or groups of 8.

In the limitations the authors of the study expressed my concerns with:

-Limited small and varied sample size

-Repeated measure trial and randomised controlled trial are limited in that sample size, setting and assessment method were all varied. Different models were used for different groups limiting the ability to make comparisons between the two groups.

Reviewer #2: This study covers a very important topic, which is very relevant especially now in these pandemic times. More than ever before, there is a need to explore ways of teaching students in the health care space, where face to face interaction is not possible. I think this is addressed well by this study. There are only a few minor corrections that I would suggest be addressed, if the study is to be published.

The Introduction is very well written, clear and concise and aims clearly outlined. I struggled a little to follow the methodology and had to go over it multiple times, and I think this is a result of the use of multiple study designs used. In particular, line 121-125 was difficult to follow. Overall the methodology was difficult to follow and in some places ambiguous. I would suggest a more concise and clearer explanation of the various groups of students and where each group was assigned. This would make the study easier to read and for the readers to follow how data was collected, and conclusions drawn. Table 1 in the results sort of makes the different groups clear to understand, but this is not explains as well in the written explanation.

6. PLOS authors have the option to publish the peer review history of their article (what does this mean?). If published, this will include your full peer review and any attached files.

Reviewer #1: No

Reviewer #2: No

---

## [Author Response · Author response to Decision Letter 0]

11 Nov 2021

Thank you for your suggestions for our manuscript, particularly regarding moving Table 1 to earlier in the manuscript as this appears to signpost the methodology far more clearly. We also are grateful to the reviewers for their valuable contributions and positive feedback. We appreciate the comments and recommendations. Please find a detailed response to suggestions below. 

Reviewer #1

Comment 1: The aim of the study was to evaluate the use of 3D printed foot models on podiatry students anxiety and confidence when using scalpels and to determine if using 3D printed models had any benefit over standard teaching practices.

The paper is well written and structured, however a few questions arise.

- In the repeated measure trial and randomised controlled trail, sample size is varied and not kept constant. Is there a reason for not using equal sample size?

Response 1: Thank you for your positive feedback and considered concerns. 

We agree that, ideally, the sample size would be more consistent between both groups and cohorts. However, recruitment was limited to the number of students enrolled in the relevant subjects at the time of the study who were willing and available to participate. The participation rate for the 2nd year cohorts in 2019 (n = 27) and 2020 (n = 24) reflects >95% of enrolled students respectively, whereas the participant rate for our 4th years in 2019 (n = 15) reflects 50% of the cohort, which was the same number who were not on external placements (and therefore unavailable) at the time of the workshop. We aimed to indicate this across the manuscript, specifically: 

 (line 123, p 6): 

“Participants were sought via purposive sampling of undergraduate podiatry students enrolled at the University of South Australia in the second year of the course in 2019 and 2020, and final year students enrolled in 2019.” 

 (line 268, p 12):

“Due to the recruitment requirements limiting the recruitment sample to students enrolled in undergraduate podiatry courses during 2019 or 2020, and a novel intervention where effect size could not be assumed, an a priori sample size calculation was not conducted.” 

However, to ensure clarity for the readers, the following has been added to Limitations section (line 511, p 22):

“Furthermore, sample sizes, limited by available student numbers, were not consistent between the year or cohort groups.”

Comment 2: -On procedure, 4th years n =15 used models on fig 2 and 3 and worked in pairs, whereas 2nd years n =24 used model on figure 1 and worked in groups of about 8. Is there a reason why the two groups did not work on the same models as well as keeping consistency of the setting and either working in pairs or groups of 8.

Response 2: The difference in procedure and models were predominantly pragmatic decisions based on need, cost and the requirements of the tasks. 

The procedure differed between the year levels (e.g., 2nd and 4th years) because 4th years are tasked with measuring, cleaning and dressing the ulcer (once they’d debrided it), requiring them to use both hands – as the models were not mounted to brackets at that time, they needed a second person to hold the model during practice. The ‘holder’ also acted as an assistant, recording measurements etc., As simple scalpel debridement only requires the use of one hand, 2nd year students could hold the model themselves. The 2nd years also need direct training, so they worked in groups of 8 so they were adequately supervised, however, each student had their own model to work on. 

The models differed between the year groups for pragmatic reasons, as the 4th years version (foot ulcer models) are printed in ninja flex, a softer flexible filament that allows the models to move more like a foot (e.g., they can put pressure against the toes and the models flex), whereas 2nd year students have a more solid model (standard PLA filament) that doesn’t move, and is far more robust so can withstand their novice scalpel ‘mishaps’ easily. In short, Ninja flex offers more lifelike models, but the extra cost and replacement requirements means it is not prudent to use them for early scalpel skill training. 

To ensure readers understand this, the rationale behind using two different models is identified in line 153 (p 8):

“There were two versions of 3D printed foot models used. A flexible Foot ulcer model with appliable lesions for final year students to use in foot ulcer management training (Fig 1), and a more rigid, robust Callused foot model for second year students learning scalpel skills for the first time (Fig 2).” 

And line 159 (p 8)

“The Foot ulcer models are printed in Ninja flex® filament to have adequate flex (to mimic foot motion) whilst the Callused 3D models are printed in standard nylon filament to be sufficiently robust (to withstand efforts from novice scalpel users).”

Whereas the difference in protocol has been updated to clarify the need for differences, (line 210, p 10):

“Students worked in pairs, helping to stabilise the model and record measures for each other, and were instructed to observe and adhere to aseptic techniques and infection control guidelines.”

And (line 215, p 10)

“Second-year students (n = 24) were introduced to the Callused 3D foot models (Fig 1) for scalpel skills teaching in small groups (n = ~8) in April 2020, each receiving their own model and given a 1-hour training session with an experienced clinical tutor. They were then allowed unlimited self-paced practice over a six-week block.”

We think this reads more clearly now and hope it satisfies your concerns. 

Comment 3: In the limitations the authors of the study expressed my concerns with:

-Limited small and varied sample size

-Repeated measure trial and randomised controlled trial are limited in that sample size, setting and assessment method were all varied. Different models were used for different groups limiting the ability to make comparisons between the two groups. 

Response 3: Thank you for these suggestions, we have altered the limitations section as below and hope this reflects your comments accordingly (line 506 to 512, p 22).

“Comparisons between the quantitative studies were limited by methodology and protocol differences, and the focus group also had a small sample size. Specifically, as the repeated measures methodology lacked control group comparisons, we are unable to determine if the measured anxiety and confidence improvements were due to the foot model exposure or related to time alone and comparisons cannot be made between this and the RCT findings. Different models and protocols were used between the second year and final year cohorts also limiting the ability to make comparisons between the cohorts. Furthermore, sample sizes, limited by available student numbers, were not consistent between the year or cohort groups.”

We thank Reviewer 1 for their input and hope we our responses have satisfied the concerns raised.

Reviewer #2: 

Comment 1: This study covers a very important topic, which is very relevant especially now in these pandemic times. More than ever before, there is a need to explore ways of teaching students in the health care space, where face to face interaction is not possible. I think this is addressed well by this study. There are only a few minor corrections that I would suggest be addressed, if the study is to be published.

The Introduction is very well written, clear and concise and aims clearly outlined. I struggled a little to follow the methodology and had to go over it multiple times, and I think this is a result of the use of multiple study designs used. In particular, line 121-125 was difficult to follow. Overall the methodology was difficult to follow and in some places ambiguous. I would suggest a more concise and clearer explanation of the various groups of students and where each group was assigned. This would make the study easier to read and for the readers to follow how data was collected, and conclusions drawn. Table 1 in the results sort of makes the different groups clear to understand, but this is not explained as well in the written explanation.

Response 1: 

We thank the reviewer for these positive comments and agree that the use of multiple study designs has increased the difficulty in keeping the methodology clear. 

In response to these concerns, and at the Editor’s suggestion, we have moved Table 1 to earlier in the article. We believe the table more clearly signposts the studies for the reader (line 115, p 6). We specifically added text to the lines as mentioned directly (now lines 135 to 138m p 8) to ensure clarity:

“A randomised control trial compared standard teaching (control) to teaching using a 3D printed foot model (intervention) for changes in anxiety and confidence in second year students (2019 cohort) who were using a scalpel for the first time. Students received 1-hour of training after they were randomly allocated to the control (n = 15) or intervention group (n = 12).”

Furthermore, we have trimmed and more clearly defined the year groups, the cohorts and the relevant study designs throughout. As we have done this over several areas, we have highlighted it directly to the manuscript. 

We thank Reviewer 2 for their input and hope you we have responded to your satisfaction.

---

## [Editor Report · Decision Letter 1]

1 Dec 2021

An exploration of the use of 3D printed foot models and simulated foot lesions to supplement scalpel skill training in undergraduate podiatry students: a multiple method study.

PONE-D-21-26441R1

Dear Dr. Banwell,

We’re pleased to inform you that your manuscript has been judged scientifically suitable for publication and will be formally accepted for publication once it meets all outstanding technical requirements.

Kind regards,

Oathokwa Nkomazana, MD MSC PhD

Academic Editor

PLOS ONE
---

## [Editor Report · Acceptance letter]

3 Dec 2021

PONE-D-21-26441R1 

An exploration of the use of 3D printed foot models and simulated foot lesions to supplement scalpel skill training in undergraduate podiatry students: a multiple method study. 

Dear Dr. Banwell:

I'm pleased to inform you that your manuscript has been deemed suitable for publication in PLOS ONE. Congratulations! Your manuscript is now with our production department. 

Kind regards, 

on behalf of

Dr. Oathokwa Nkomazana 

Academic Editor

PLOS ONE